# Combined impact of local climate and soil properties on soil moisture patterns

Thushara Gunda<sup>1</sup>, Udeni P. Nawagamuwa<sup>2</sup>, and George M. Hornberger<sup>1</sup>

<sup>1</sup>Vanderbilt University Institute for Energy and the Environment, Department of Civil and Environmental Engineering, Vanderbilt University, Nashville, TN, USA <sup>2</sup>Department of Civil Engineering, University of Moratuwa, Katubedda, Moratuwa, Sri Lanka (10400)

*Correspondence to:* Thushara Gunda (tgunda@gmail.com)

**Abstract.** Soil plays a key role in terrestrial water dynamics by retaining precipitation on land. A water balance approach is used to evaluate spatial and temporal variations in soil moisture in Sri Lanka, a country characterized by high spatial variability as reflected in the recognition of three regions of the country, the wet zone, the intermediate zone, and the dry zone. We show that a combination of local climate and soil properties drive spatial patterns of soil moisture deficits on the island, with soils

- buffering climate variability in the wet zone and enhancing drought patterns in the dry zone. Changes in historical temporal patterns are most notable for the intermediate zone, a region characterized by consistently variable deficits. Counterfactuals of climate change scenarios indicate temperature will drive increases in deficit likelihoods (up to 20 %) in the future, with greatest impact in the intermediate and dry zones, where more than 80 % of the national rice production is concentrated. Given that temperature projections are less uncertain than other climate change impacts, further evaluation of future water stresses
- are needed. Coupled with remotely-sensed soil moisture data, the findings from this study have implications for infrastructural planning and seasonal crop water allocations in zones with a degree of variability (i.e., neither consistently wet nor consistently dry). Because soil hydrologic regimes reflect inherent, local vulnerabilities, water management decisions need to incorporate regional variabilities in soil moisture dynamics in assessments of climate change adaptations.

Copyright statement. The authors certify that they have read and agree to the terms outlined by HESS.

## 15 1 Introduction

Soil moisture plays a key role in hydrological processes (such as runoff generation) and ecosystem functions (such as soil respiration and soil carbon and nutrient cycling) (Botter et al., 2007; Manzoni et al., 2012). In addition to influencing soil forming processes (Karmakar et al., 2016), soil moisture serves as a critical storage reserve of water for evapotranspiration processes (McColl et al., 2017). Strong relationships observed with crop yields underscore the importance of monitoring soil

moisture for food security (Holzman et al., 2014). Characterizing soil moisture variability has provided valuable insights into droughts, extreme temperatures, floods, and thunderstorms (Fournier et al., 2016; Heim Jr, 2002; Koster et al., 2004; Massari et al., 2014; McColl et al., 2017; Whan et al., 2015).

Soil moisture is influenced by both physical and social variables. Temperature directly impacts the loss of water from soil through evapotranspiration while precipitation serves as a source of water (Várallyay et al., 2010). Soil moisture is also influenced by land use patterns, which can drive differences in evapotranspiration and physical properties of soil (Fu et al., 2003). In particular, land management and irrigation practices in agricultural areas can greatly influence soil moisture dynamics (Pender and Kerr, 1998); irrigation practices may vary depending on farmers' sensitivities to changes in soil water content (Du

et al., 2017).

Assessing the potential impacts of climate change on soil moisture regimes is complex because of the strong feedbacks to both temperature and precipitation changes (Seneviratne et al., 2010; Várallyay et al., 2010). Future projections of temperature changes generally have lower uncertainties than those for precipitation (IPCC, 2014). Mean surface temperatures are expected

- to increase between 1.1 and 4.8C, with greater impacts at higher latitudes; precipitation changes are much more variable both geographically and seasonally, making quantitative estimates more uncertain (IPCC, 2014). Societal impacts of climate change are expected to vary globally, with agricultural production in regions at lower latitudes disproportionally impacted due to low adaptive capacity and already high temperatures (Gunda et al., 2017). Given that crop productivity is strongly impacted by water availability (Kang et al., 2009), understanding potential impacts of climate change on soil moisture is particularly
- important for development of informed adaptation strategies.

Although studies have quantified impacts of climate change on soil moisture at both global (e.g., Berg et al. (2017)) and regional (e.g., Eitzinger et al. (2003)) scales, limited attention has been given to characterizing differences in soil moisture patterns arising from variabilities in local climate and soil conditions. Research shows that evapotranspiration at the scale of a watershed is related to climate drivers and to soil capacity to retain water in complex ways (Garcia and Tague, 2015) and that

long-term patterns in soil moisture reflect climate interactions with soil water retention capacity (Salley et al., 2016). Because soil properties are intrinsic characteristics of local systems, societal responses may need to account for regional variabilities in soil moisture patterns in planning efforts.

In our work, we explore how climate and soil interact using both historical climate observations and stylized climate change scenarios. Specifically, we aim to assess how spatial variations in climate and soil properties influenced patterns of soil moisture over the last century and how spatial and seasonal variations of soil moisture may change in the future due to climate change.

- over the last century and how spatial and seasonal variations of soil moisture may change in the future due to climate change. We use a water balance approach in conjunction with counterfactuals to address the research questions; satellite data is used to evaluate the general patterns identified by the water balance approach. The island nation of Sri Lanka is largely agricultural with high spatial and temporal rainfall variability (Gunda et al., 2016). We use this area for analysis as it has the high spatial variability characteristics that underlie our questions. Our results suggest that spatial differences in soil moisture variability
- are driven by a combination of climate and soils, with the latter buffering the magnitude and timing of deficits during wet conditions. Both historical patterns and climate change impacts exhibit spatial variability, with deficits in the zones with a degree of variability (i.e., neither consistently wet nor consistently dry) most likely to be amplified in the future.

# 2 Methods

#### 2.1 Study site

Located off the southeast coast of India, the island of Sri Lanka experiences a tropical climate. The physiography of Sri Lanka consists of a sloping topography with high country (i.e., elevation > 1000 m) in the center of the island that drops to less

- than 100 m along the coast (Figure 1). The average temperature on the island is 27 C except in the central highlands, where temperatures are 15 C. Annual precipitation ranges between 1000 and 3600 mm on the island, with the wet zone receiving more than 2500 mm of rainfall and the dry zone receiving less than 1750 mm (Zubair, 2003). The rainfall differences arise from the southwest portion of the island receiving rainfall during both the southwest monsoon (May-Sept) and the northeast monsoon (Dec-Feb) while the dry zone only receives rainfall during the latter; the intermediate zone is a transition region (Gunda et al.,
- 2016). In addition to rainfall, the three zones are also demarcated by varying soil types, with latosols and regosols (LRs) being most prevalent in the wet zone, reddish brown earths (RBEs) and red latosols (RLs) in the dry zone, and a mixture of soils in the intermediate zone; RLs underlie the northwest coastal regions of the island of Puttalam and Jaffna. LRs are characterized by high water holding capacities while RBEs and RLs have low water holding capacities (Moorman and Panabokke, 1961; Cooray, 1984). RBEs and RLs are also characterized by rapid infiltration and rapid releases of soil moisture at low tensions
- (DOA, 2017). Rice production (the staple food of the country) occurs predominantly in the intermediate and dry zone districts of the country (Davis et al., 2016). The limited water supplies during the period from April to August are supplemented by local reservoirs in the intermediate zone and by the Mahaweli irrigation system, which diverts water from the central highlands to the lowlands, in the dry zone. Sri Lanka has been self-sufficient in rice production since 2005 but elevated temperatures and shifting precipitation patterns are sources of crop stress for the future (Davis et al., 2016).

## 20 2.1.1 Soil moisture calculations

Strong correlation between Palmer Drought Severity Index (PDSI)-derived soil moisture and independent soil moisture estimates have been observed globally (Dai et al., 2004; Szép et al., 2005). A common measure of agricultural drought, PDSI uses a physical water balance in a 2-layered, 1-m soil system (Palmer, 1965). Precipitation and recharge are inputs into the soil system while evapotranspiration and runoff are outputs. Runoff is dependent on the saturation of the underlying soils, which

are characterized by the available water content (AWC) and antecedent conditions. The time scale of PDSI is approximately 9 months (Heim Jr, 2002). A calculation of soil moisture content as part of the balance allows use of PDSI as a soil moisture indicator (Dai et al., 2004; Szép et al., 2005). The primary advantage of using a drought index approach is that it can leverage long temporal records of meteorological data (Seneviratne et al., 2010).

Monthly soil moisture was calculated at 13 stations in Sri Lanka (Figure 1), using the PDSI tool provided by Jacobi et al. 30 (2013) with the Thornthwaite method for calculating potential evapotranspiration values and the full period of record for calibration. Long-term, monthly precipitation and average temperature data for the 13 stations, were obtained from the Meteorological Department of Sri Lanka and processed as outlined in Gunda et al. (2016); the 13 stations capture the climate spatial variabilities of the country (Gunda et al., 2016). PDSI was calculated on data from 1875 to 2016, with stable values achieved

**Figure 1.** Stations with long-term historical data in Sri Lanka. (a) General location of Sri Lanka, (b) local topography with station locations, and (c) zone boundaries with station locations, with marker size indicating the local available water content (AWC) of soils. Water is captured in the high elevation regions of the country and transferred to the lowlands to the north and southeast for agricultural production.

from Jan 1878 to Mar 2014. AWC values were derived from the literature and range from 37 mm in the northwest coastal regions to 187 mm in the wet zone (Aydin et al., 2012; De Silva and Rushton, 2007; Mikunthan and De Silva, 2010; Keerthisena et al., 2001; Mapa et al., 2010; Rajapaksha et al., 2002) (Figure 1). Soil moisture deficits (SMD), a common metric for climate change analysis (Cohen et al., 1996), were estimated by subtracting soil moisture values from the corresponding AWC values.

5 The SMDs were normalized by AWCs to represent the fraction of soil moisture deficit relative to the local available water content for the top 1 m (Peng et al., 2017).

SMD estimates at the 13 stations were evaluated as a function of zone, AWC, and season to understand spatial and temporal patterns of variability. Median and standard deviation SMD values were calculated for each month to explore both the magnitude and timing of deficits. Seasonal patterns were also evaluated using soil moisture data observed directly by the Soil

10 Moisture Active Passive (SMAP) mission, which was launched in January 2015. SMAP measures volumetric soil moisture in the top 5 cm of the soil (Sun et al., 2017). A Level 3, 36-km gridded product (L3\_SM\_P) derived from the passive sensor was used to develop monthly estimates of soil moisture. Grid values were extracted for the 13 station locations; cells with high percentage of water bodies and excessive vegetation were excluded from analysis. Temporal patterns were characterized by calculating exceedance probabilities of mean annual deficits, both for the full record and for the record split into two periods

("Until 1946" and "After 1946") to evaluate changes in the historical record. The nonparametric Wilcoxon-Mann-Whitney test was used to assess differences in the two period probability curves (Conover and Iman, 1981).

## 2.1.2 Climate scenario analysis

- Both elevated temperatures and shifting precipitation patterns can impact SMDs. Climate change projections for Sri Lanka note 5 consistent increases in temperature throughout the year while precipitation shifts are variable based on the season (De Silva et al., 2007; Zubair et al., 2015). The largest modes of precipitation variability coincide with periods of high rainfall, notably during Apr-Jun and Oct-Dec (Zubair et al., 2015). Some model projections show increases in rainfall while others show decreases for the future (Seo et al., 2005). The large range of projections is driven by complex monsoon dynamics in the Indian Ocean that lead to large biases in the climate models (Li et al., 2015).
- 10 To assess sensitivity of SMDs to elevated temperatures, we consider warming scenarios of 1C, 2C, and 3C to account for the range of conditions projected for Sri Lanka (Zubair et al., 2015). For each warming scenario, we consider the "worst case scenario" of an added impact of a 10 % decrease in precipitation during the high rainfall months of Apr-Jun and Oct-Dec aforementioned. Because antecedent conditions are incorporated in the PDSI calculations, the sequence of meteorological occurrences are important. Therefore, the sensitivity of SMDs to elevated temperatures and reduced precipitation were evaluated
- using a counterfactual approach, whereby we impose the climate change patterns on the historical data set. In other words, the historical meteorological data at each station was modified by climate change patterns (i.e., warming only or warming and seasonal precipitation decreases) and then inputted into the PDSI tool to estimate SMDs and evaluate how soil moisture deficit patterns could change in the future. This approach would limit overestimation issues since climate change projections imposed on the record are dynamically accounted for in the PDSI calculations, rather than being imposed on mean patterns (Berg et al., 2017).
- 20 2017).

### 3 Results

Deficits are generally lowest in the wet zone and highest in the dry zone, with the intermediate zone experiencing consistently high variability in monthly values (Figure 2). There is little intra-zone variability in the intermediate and dry zones; the wet zone stations of Nuwara Eliya and Ratnapura have much lower deficits and associated variabilities than those observed at the

other two wet zone stations (Figure A1). Jan-Mar deficits in the wet zone are driven by the coastal stations, Colombo and Galle, which have high standard deviations throughout the year (Figures 3 and A2). High deficits (i.e., fraction of AWC > 0.7) are present Jul-Sep in the intermediate zone and Feb-Sep in the dry zone, with high variability Oct-Jan and low variability May-Aug in the dry zone (Figures 3 and A2).

SMAP data show higher soil moisture in Colombo and low soil moisture Jan-Mar and Jul-Sep at all three stations (Figure 4).
During the first six months of the year, satellite data indicate peak surface soil moisture in May at Kurunegala and Anuradhapura while water balance data indicate lowest deficits in April in the top 1 m of soil at those locations (Figure 4). Exceedance probability curves of the three zones highlight the prevalence of deficits in the dry zone, with the intermediate zone observing