# Peer review of "Combined impact of local climate and soil properties on soil moisture patterns"

_Hydrology and Earth System Sciences, 2017_

## Referee Comment (RC1) · Anonymous Referee #1 · 3 Aug 2017

The manuscript by Gunda et al., (2017) evaluated the soil moisture patterns in Sri Lanka based on soil moisture deficient that was calculated from meteorological measurements. The soil moisture from SMAP was also applied for the analysis. In general, the study was well designed and written. However, I would like to discuss the following points with the authors:

1. The title of the paper emphasizes the soil moisture patterns. And the spatial variations of soil moisture were highlighted in the whole manuscript such as abstract, introduction, discussion and conclusion sections. I expected that there are figures or plots showing spatial patterns of the soil moisture. However, the current analyses are mainly based on station measurements. I think it is not appropriate to use spatial patterns or spatial variations in the manuscript.

[Figure]

2. SMAP soil moisture was used to cross compare with the calculated soil moisture deficits such as Figure 4. In page 8 line 1, the statement is not true, because the seasonal patterns from SMAP and calculated deficit are quite different for station Colombo. The authors should discuss the reasons. In addition, I am wondering how accurate is SMAP soil moisture over these stations. Did you validate the SMAP soil moisture over the study area? Why do you choose SMAP rather than other soil moisture products such as ESA CCI?

3. The name of the sections should be more precise, for example, section 2 Methods should be Study area and methods. Section 2.1 should be Study area rather Study site. Section 2.1.1 Soil moisture calculations should be Soil moisture deficit calculation. Could you provide more details on the equations of soil moisture deficit? It is not clear for me how do you calculate SMD.

4. The conclusion part could be improved to answer the research questions that were raised in the introduction section. In addition, the last sentence is not clear for me, as far as I know, the downscaling method for SMAP can only provide soil moisture at 1-10 km. For regional agriculture applications, the soil moisture at tens of meters is more required. Is soil moisture at scale of km enough for the "informing local practices" ?

---

## Author Comment (AC1) · 31 Aug 2017

Responses to Referee # 1 Comments (latter are bulleted)
The manuscript by Gunda et al., (2017) evaluated the soil moisture patterns in Sri Lanka based on soil moisture deficient that was calculated from meteorological measurements. The soil moisture from SMAP was also applied for the analysis. In general, the study was well designed and written. However, I would like to discuss the following points with the authors:
Response: We thank the referee for a thoughtful assessment of this paper. We have made modifications, as summarized below, in response to the points raised by the referee and are confident that this has resulted in a significantly improved manuscript.

1. The title of the paper emphasizes the soil moisture patterns. And the spatial variations of soil moisture were highlighted in the whole manuscript such as abstract, introduction, discussion and conclusion sections. I expected that there are figures or plots showing spatial patterns of the soil moisture. However, the current analyses are mainly based on station measurements. I think it is not appropriate to use spatial patterns or spatial variations in the manuscript.

Response: Since the focus of our analysis is long-term changes in soil moisture patterns and satellite measurements only provide spatial coverage for recent decades, we used station-level measurements, which are available from 1880s in Sri Lanka. Station-level measurements have been successfully used to understand variabilities in space for soil moisture analyses (e.g., Hirschi et al., 2011, Nature Geoscience, doi: 10.1038/ngeo1032). Although the stations we used capture the dominant spatial variabilities in climate across Sri Lanka, we recognize that they are not capable of capturing the variabilities across soil types. This is one of the primary reasons that our analysis focuses on spatial patterns at a zonal-scale (i.e., the wet, intermediate, and dry zones) and a station-scale. However, in acknowledgement of the reviewer's point regarding the value of country-wide visuals, we will add supplementary material that presents the interpolated historical soil moisture deficit values over space and time.

2. SMAP soil moisture was used to cross compare with the calculated soil moisture deficits such as Figure 4. In page 8 line 1, the statement is not true, because the seasonal patterns from SMAP and calculated deficit are quite different for station Colombo. The authors should discuss the reasons. In addition, I am wondering how accurate is SMAP soil moisture over these stations. Did you validate the SMAP soil moisture over the study area? Why do you choose SMAP rather than other soil moisture products such as ESA CCI?
Response: The seasonal patterns in Colombo from SMD vs SMAP show generally good agreement, with high soil moisture during May-Jun and Nov-Dec and low soil moisture during Aug-Sep. There is some disagreement between the two datasets regarding the timing of low soil moisture values in the first half of the year, with SMD indicating lowest values in March while SMAP indicating lowest values in January. These differences are most likely due to the different time periods covered by the two datasets; SMD data covers 1878-2014 while SMAP data covers 2015-2017. The discrepancy in these patterns reflects a rainfall shift over the last few years (https://www.worldweatheronline.com/colombo-weatheraverages/western/lk.aspx), and correspondingly, shift in soil moisture deficit towards earlier months. We will update the discussion to include this information. Regarding validation, we were not able to directly validate SMAP data over the study area due to lack of direct soil moisture measurements. However, the accuracy of SMAP data has been assessed across multiple regions of the world (Chan et al., 2016, IEEE Transactions on Geoscience and Remote Sensing, doi: 10.1109/TGRS.2016.2561938; Colliander et al., 2017, Remote Sensing of Environment, doi: 10.1016/j.rse.2017.01.021) and we followed the quality protocol associated with the dataset to exclude poor signal qualities (associated with heavily forested and high water regions) from our analysis. In addition to comparing general seasonal patterns, one of our goals for using the SMAP dataset is to highlight future capabilities in soil moisture monitoring. A significant advantage of the SMAP dataset over other soil moisture products is the near real-time availability of this dataset (it is updated every 2-3 days), which can be incredibly helpful for local planning efforts.

3. The name of the sections should be more precise, for example, section 2 Methods should be Study area and methods. Section 2.1 should be Study area rather Study site. Section 2.1.1 Soil moisture calculations should be Soil moisture deficit calculation. Could you provide more details on the equations of soil moisture deficit? It is not clear for me how do you calculate SMD.

**HESSD**
Response: The water balance data generated at each station (as part of the PDSI calculations) includes soil moisture values. The SMD was calculated by: 1) subtracting these soil moisture values from the corresponding station's available water content (AWCs; i.e., max soil moisture) and 2) normalizing by the AWCs. The resulting SMD then represents the fraction of soil moisture deficit relative to the local available water content for the top 1 m. We will add an equation and reorganize some of the text to clarify the calculations associated with the SMD. We will also revise the section names as suggested.

4. The conclusion part could be improved to answer the research questions that were raised in the introduction section. In addition, the last sentence is not clear for me, as far as I know, the downscaling method for SMAP can only provide soil moisture at 1-10 km. For regional agriculture applications, the soil moisture at tens of meters is more required. Is soil moisture at scale of km enough for the "informing local practices"?

Response: By local, we were referring to national-level and zonal-level agricultural planning efforts (i.e., not farm-level analyses), which can be greatly informed by down-scaled soil moisture datasets. We will revise the conclusion to clarify our intentions.

---

## Referee Comment (RC2) · Anonymous Referee #2 · 11 Sep 2017

The study by Gunda et al. aimed to evaluate the impacts of climate and soil properties on long-term soil moisture patterns in Sir Lanka. The authors adopted a simple water balance model to compute monthly deficits of soil moisture under different climatic conditions. Overall, the manuscript was well written and reasonably structured. Although this study presents some interesting model results, in my opinion, it suffers from some major problems and the significance of this research is not clear to me.

(1) The authors divided the study region into three climate zones (e.g., wet, intermediate, and dry). However, all those zones belong to humid tropic climates. Therefore, the authors need to stress those are relative terms.

(2) To be a standalone paper, the authors need to provide the description of the water balance model used by them as well as all model parameters. My major concern is

the use of the simple water balance model, which does not seem appropriate for the aim of this study. Except for the use of SMAP data, it lacks rigorous model validations. In addition, the use of a 1-D model is not justified by the significant slope across the study region. It would be very important to provide information relevant to the current study, such as soil properties, observed soil moisture, groundwater level, and climate (e.g., precipitation, potential evapotranspiration, and their seasonality), which is critical for the interpretation of the data and to ensure the validity of the model results.

(3) The use of the SMAP data is not appropriate for several obvious reasons (e.g., mismatch in spatial scales and soil depth, and a very limited time period). More importantly, from Figure 4, we can see that when the deficit fraction reached above 0.75, SMAP data still showed very high soil moisture contents (deeper soil moisture tended to be even higher). Physically, this does not make any sense.

(4) The authors argued that soil had a buffering effect in the wet zone, and temperature had a larger effect than precipitation on the shifts in soil moisture patterns. On the other hand, in energy limited environments as in the study region, available energy is of course more important. It simply might be that there is not enough energy to evaporate soil moisture in those areas, despite increases in temperature.

Overall, I think the authors need to provide more information and data to justify the use of the water balance model and the validity of the model results.

Minor comments:

(1) P3L5: it should be degree Celsius not C.

(2) P3L21: either use 'correlations' or change 'have' to 'has'.

(3) P3L22: 'As a common measure...'

(4) P4L1: 'ranged...'

(5) P4L8: '... standard deviation of ...'

(6) P5L29: 'in Jan-Mar...' (and in other places as well)

(7) P7L1: '... conditioned both on ...'

(8) P10L4: Is that possible the lag time is the artifact of the model structure?

---

## Author Comment (AC2) · 14 Sep 2017

The study by Gunda et al. aimed to evaluate the impacts of climate and soil properties on long-term soil moisture patterns in Sir Lanka. The authors adopted a simple water balance model to compute monthly deficits of soil moisture under different climatic conditions. Overall, the manuscript was well written and reasonably structured. Although this study presents some interesting model results, in my opinion, it suffers from some major problems and the significance of this research is not clear to me.

Response: We thank the referee for a thoughtful assessment of this paper. We have made modifications, as summarized below, in response to the points raised by the referee and are confident that this has resulted in a significantly improved manuscript.

(1) The authors divided the study region into three climate zones (e.g., wet, intermediate, and dry). However, all those zones belong to humid tropic climates. Therefore, the authors need to stress those are relative terms.

Response: In acknowledgement of the referee's observation, we will add sentences in both the methods and discussion sections to clarify that the zone names (i.e., wet, intermediate, and dry) indicate relative conditions, although they broadly correspond with zones with different Koppen climate classifications.

(2) To be a standalone paper, the authors need to provide the description of the water balance model used by them as well as all model parameters. My major concern is the use of the simple water balance model, which does not seem appropriate for the aim of this study. Except for the use of SMAP data, it lacks rigorous model validations. In addition, the use of a 1-D model is not justified by the significant slope across the study region. It would be very important to provide information relevant to the current study, such as soil properties, observed soil moisture, groundwater level, and climate (e.g., precipitation, potential evapotranspiration, and their seasonality), which is critical for the interpretation of the data and to ensure the validity of the model results.

Response: PDSI has been effectively used to understand drought patterns throughout the globe (Dai, 2013, Nature Climate Change, doi: 10.1038/nclimate1633; Vicente-Serrano et al., 2010, Journal of Hydrometeorology, doi: 10.1175/2010JHM1224.1). In particular, we note in the text that PDSI-derived soil moisture have shown strong correlations with independent soil moisture estimates throughout the globe (Szép et al., 2005, Physics and Chemistry of the Earth, Parts A/B/C, doi: 10.1016/j.pce.2004.08.039). The PDSI model conducts a physical water balance of precipitation, evapotranspiration, recharge, and runoff dynamics, assuming a twolayered, one-meter soil system. Precipitation occurring in a given month is first utilized to meet the evapotranspiration demand of that month; if a given month's precipitation is higher than that month's evapotranspiration demand, then there is a positive moisture anomaly and vice versa. During positive moisture anomalies, moisture is transferred to the top layer until it reaches saturation, and then transferred to the bottom layer. When both layers are saturated, excess water becomes runoff. Although the PDSI uses a conceptually-simple water balance model approach, the associated calculations, which account for antecedent conditions, are quite complex. The PDSI Matlab tool we used in the analysis was developed by Jacobi et al., (2013; Water Resources Research, doi: 10.1002/wrcr.20342), who outline the multiple functions and accounting measures associated with PDSI calculations. Additional information regarding the methodology and limitations of PDSI are provided in Palmer (1965; US Department of Commerce Research Paper no. 45), Alley (1984; Journal of Climate and Applied Meteorology, doi: 10.1175/1520-045), and Briffa et al. (1994; International Journal of Climatology, doi:10.1002/joc.3370140502). The model parameters are: 1) monthly precipitation and temperature values, 2) station latitudes, and 3) available water content (AWC) of soils. The PDSI model uses the station latitudes in combination with the monthly temperature values to calculate a given month's evapotranspiration (ET) demand; we used the Thornthwaite model to calculate the potential ET values. The use of temperature-based ET models in PDSI calculations could lead to overestimation issues (Sheffield et al., 2012, Nature, doi: 10.1038/nature11575). However, it has been well recognized that insights generated from PDSI could still be useful as long as its shortcomings are recognized (Trenberth et al., 2014, Nature Climate Change, doi: 10.1038/nclimate2067). We will add information about seasonal temperature and precipitation patterns in the three zones, including a new supplementary figure, which will be referenced in the site description section. We do not have information regarding observed soil moisture and groundwater level conditions in Sri Lanka but information about the local soil properties is provided in Figure 1. We will update the text to reflect the strengths and drawbacks associated with PDSI mentioned above. We will

also revise the Methods section to include details about the model calculations and associated parameters noted above.

(3) The use of the SMAP data is not appropriate for several obvious reasons (e.g., mismatch in spatial scales and soil depth, and a very limited time period). More importantly, from Figure 4, we can see that when the deficit fraction reached above 0.75, SMAP data still showed very high soil moisture contents (deeper soil moisture tended to be even higher). Physically, this does not make any sense.

Response: Our primary motivation for using SMAP is to provide a general comparison of seasonal patterns for the PDSI-derived soil moisture values and not to validate the PDSI-derived SMD data. Although SMAP only focuses on the top 5 cm while SMD values account for the top 1m of soil, studies (e.g., Berg et al., 2017, Geophysical Research Letters, doi: 10.1002/2016GL071921) indicate that moisture dynamics between the surface and total unsaturated zone generally do agree. Indeed, that is what we observe in our data. For example, whenever the deficit fraction is greater than 0.75, SMD values are generally low, with values less than 0.3 cm3/cm3 at the intermediate zone station and less than 0.25 cm3/cm3 at the dry zone station. Although both soil moisture datasets are fractions, they are calculated differently; one is normalized to volume of soil (SMAP) while the other to volume of water (deficit fraction). So to facilitate an easier comparison of the general seasonal patterns, we inverted the axis of Figure 4a. There is some disagreement between the two datasets regarding the timing of low soil moisture values in the first half of the year at Colombo, with SMD indicating lowest values in March while SMAP indicating lowest values in January. These differences are most likely due to the different time periods covered by the two datasets; SMD data covers 1878-2014 while SMAP data covers 2015-2017. The discrepancy in these patterns reflects a rainfall shift over the last few years (https://www.worldweatheronline.com/colombo-weather-averages/western/lk.aspx), and correspondingly, shift in soil moisture deficit towards earlier months. In addition to comparing general seasonal patterns, one of our goals

for using the SMAP dataset is to highlight future capabilities in soil moisture monitoring. A significant advantage of the SMAP dataset over other soil moisture products is the near real-time availability of this dataset (it is updated every 2-3 days), which can be incredibly helpful for national- and zonal-level planning efforts. We mention in the discussion that continued development of statistical downscaling methods could enable use of SMAP data to a scale usable for even more localized water management efforts. We will update the text to clarify the motivation for using SMAP data and revise the discussion to include information about observed discrepancies between the two datasets mentioned above.

(4) The authors argued that soil had a buffering effect in the wet zone, and temperature had a larger effect than precipitation on the shifts in soil moisture patterns. On the other hand, in energy limited environments as in the study region, available energy is of course more important. It simply might be that there is not enough energy to evaporate soil moisture in those areas, despite increases in temperature.

Response: We agree with the referee's observations. Increasing temperatures does provide more energy, which could enhance soil evaporation in an energy-limited environment. In fact, the general lack of soil moisture deficits in the wet zone does indicate an energy-limited environment. The buffering effect of soils refers to the timing of the deficits. Under the climate change scenarios, the wet zone experienced deficits a few months after the precipitation decreases indicating that the local soils (which have higher water holding capacities) did not immediately respond to changes in the local climate. The dry zone, on the other hand, with its high prevalence of deficits is indicative of a water-limited environment. The soils of the dry zone could not provide a buffer against local climate changes due to their low AWC values. We have revised the discussion to include information regarding energy-limited and water-limited environments.

Overall, I think the authors need to provide more information and data to justify the use of the water balance model and the validity of the model results.

Response: We thank the referee for a thoughtful assessment of this paper. We believe the modifications summarized above will provide the justifications for the water balance model use and its results.

Minor comments: (1) P3L5: it should be degree Celsius not C.

Response: We will change all notations of C to °C

(2) P3L21: either use 'correlations' or change 'have' to 'has'.

Response: We will change the word to "correlations"

(3) P3L22: 'As a common measure. . .'

Response: For clarity, we will revise the sentence to "PDSI is a common measure of agricultural drought and uses . . ."

(4) P4L1: 'ranged. . .'

Response: The AWC values of soils represent present conditions so we maintained the present tense in the text.

(5) P4L8: '. . . standard deviation of . . .'

Response: We have added the word "of" in the sentence

(6) P5L29: 'in Jan-Mar. . .' (and in other places as well)

Response: We updated this text to clarify "the periods of Jan-Mar and Jul-Sep. . ."

(7) P7L1: '. . . conditioned both on . . .'

Response: We revised the sentence to clarify that the soil moisture deficit is "influenced by" both the soils water holding capacities and climate conditions.

(8) P10L4: Is that possible the lag time is the artifact of the model structure?

Response: The time scale of PDSI is approximately 9 months (Heim Jr, 2002, Bulletin

of the American Meteorological Society, doi: 10.1175/1520-0477(2002)083). There-fore, the 3-month lags we observe in our scenario results are most likely not an artifact of the model structure.